



# Quantifying stocks in exchangeable base cations in permafrost: a reserve of nutrients about to thaw

Elisabeth Mauclet[1], Maëlle Villani[1*], Arthur Monhonval[1], Catherine Hirst[1], Edward A. G. Schuur[2], Sophie Opfergelt[1]

[1]Earth and Life Institute, Université catholique de Louvain, Louvain-la-Neuve, Belgium
[2]Center for Ecosystem Society and Science, Northern Arizona University, Flagstaff, AZ, USA

*Correspondence to*: Maëlle Villani (maelle.villani@uclouvain.be)

**Abstract.** Permafrost ecosystems are limited in nutrients for vegetation development and constrain the biological activity to the active layer. Upon Arctic warming, permafrost degradation exposes large amounts of soil organic carbon (SOC) to

decomposition and minerals to weathering, but also releases organic and mineral soil material that may directly influence the soil exchange properties (cation exchange capacity and base saturation). The soil exchange properties are key for nutrient base cation supply ($Ca^{2+}$, $K^+$, $Mg^{2+}$) for vegetation growth and development. In this study, we investigate the distribution of soil exchange properties within typical Arctic tundra permafrost soils at Eight Mile Lake (Interior Alaska, USA) because they will dictate the potential reservoir of newly thawed nutrients and thereby influence soil biological activity and vegetation nutrient

sources. Our results highlight a difference in the SOC distribution within soil profiles according to the permafrost thaw. The poorly thawed permafrost soils (active layer thickness; ALT ≤ 60 cm) present more organic material in surface (i.e., organic layer thickness; OLT ≥ 40 cm) than the highly thawed permafrost soil (i.e., ALT > 60 cm and OLT < 40 cm). In turn, this difference in SOC distribution directly affects the soil exchange complex properties. However, the low bulk density of organic-rich soil layers leads to much lower CEC density in surface (~9 400 $cmol_c$ $m^{-3}$) than in the mineral horizons of the active layer

(~16 000 $cmol_c$ $m^{-3}$) and in permafrost soil horizons (~12 000 $cmol_c$ $m^{-3}$). As a result of the overall increase in CEC density with depth and the overall increase in base saturation with depth (from ~20% in organic surface to 65% in permafrost soil horizons), the average total stock in exchangeable base cations ($Ca^{2+}$, $K^+$, $Mg^{2+}$ and $Na^+$ in g $m^{-3}$) is more than 2-times higher in the permafrost than in the active layer. More specifically, the stocks in base cations in the upper part of permafrost about to thaw in the following are ~ 860 g $m^{-3}$ for $Ca_{exch}$, 45 g $m^{-3}$ for $K_{exch}$, 200 g $m^{-3}$ for $Mg_{exch}$ and 150 g $m^{-3}$ for $Na_{exch}$. This first

order estimate is a needed step for future ecosystem prediction models to provide constraint on the size of the reservoir in exchangeable nutrients (Ca, K, Mg) about to thaw.

## 1 Introduction

Northern ecosystems are characterized by low temperatures and the presence of ground underlain by permafrost. Within permafrost soils, the surface ground layer which thaws seasonally in the summer and refreezes in winter is referred to as the

active layer. Active layer thickness is crucial because it governs the soil volume for plant rooting systems, biogeochemical





activity, hydrological processes, and the amount of organic and mineral soil constituents exposed to above-freezing seasonal temperatures (Blume-Werry et al., 2019; Hinzman et al., 2003; Kane et al., 1991). In parallel, the presence of permafrost restricts downward movement of surface water and may create waterlogged soil conditions (Hinzman et al., 2003; Schuur et al., 2015). At the transition between the base of the active layer and the top of permafrost, the transient layer may contain thaw

unconformities reflecting previous periods of exceptional permafrost thaw (Shur et al., 2005). Together, cold temperatures and water-saturated conditions reduce the decomposition rates of soil organic matter (SOM) mainly originating from dead plant tissues and lead to high SOM accumulation in surface (Schuur et al., 2008; Zimov et al., 2006). Furthermore, cryogenic processes can mix soil material by repeated freezing and thawing and create larger organic inclusions at depth than previously recognized (Bockheim and Hinkel, 2007; Ping et al., 2008).


Soil organic matter shows specific properties such as low bulk density, source of acidity, large amounts of negatively charged sites at the organic compound surface (Askin and Özdemir, 2003; Périé and Ouimet, 2008; Ping et al., 2005), and thereby largely influences multiple soil physical and chemical properties. In particular, SOM and clay minerals both contribute to the soil cation exchange capacity (CEC), which is a soil parameter that evaluates the soil complex ability to retain exchangeable

cations and prevent their lixiviation or further drainage (Doran and Safley, 1997; Feller et al., 1991; Oades et al., 1989; Stevenson, 1994). Among exchangeable cations, the most common base cations $Ca^{2+}$, $K^+$, $Mg^{2+}$ and $Na^+$ act as buffer against soil acidification (Bowman et al., 2008; Tian and Niu, 2015; Ulrich, 1983). Additionally, concentration in exchangeable acid cations as $H^+$ and $Al^{3+}$ is expressed as the exchange acidity (Peverill et al., 1999). The base saturation approximates the relative proportion of base cations adsorbed onto the soil exchange complex and is related to other soil properties and external

factors as weathering stage, parent material, organic matter content, climate and vegetation. Some base cations are essential nutrients for plant growth and development ($Ca^{2+}$, $K^+$ and $Mg^{2+}$) and the exchangeable soil fraction constitutes a pool of cations readily available for plant uptake (Binkley and Vitousek, 1989; Marschner, 2012). More broadly, plant nutrient availability results from dynamic interaction of soil processes (as weathering, atmospheric deposition, leaching and biological cycling; Jobbágy and Jackson, 2001) and the surface exchange reactions are especially important for the limiting nutrient K,

but also for the key nutrients Ca and Mg (Havlin, 2005; Krull et al., 2004; Peverill et al., 1999).

Upon warming in the Arctic, permafrost degradation results in active layer thickening, ground subsidence and changing soil moisture conditions (Hirst et al., 2022; Olefeldt et al., 2016; Osterkamp, 2005; Osterkamp et al., 2009). Moreover, permafrost degradation may influence the balance between organic and mineral constituents forming the soil exchange complex. In

particular, enhanced SOM microbial degradation (Hobbie and Chapin, 1998; Nadelhoffer et al., 1992; Schuur et al., 2015; Shaver et al., 2006) or increased lateral transport of organic soil material by water fluxes (Plaza et al., 2019) generate massive loss of soil organic carbon (SOC). As a result, changes in SOC content and distribution with depth may influence the soil properties distribution. Although the SOC stock and distribution in permafrost-affected soils is well characterized (Hugelius et al., 2014; Tarnocai et al., 2009; Strauss et al 2017; Schuur et al 2018), its evolution is uncertain and partly depends on the



vulnerability of organic constituents to decompose upon soil warming and permafrost thaw (Schaefer et al., 2011; Schuur et
      al., 2008). Additionally, permafrost thaw exposes soil material at depth (Beermann et al., 2017; Keuper et al., 2017; Salmon
      et al., 2016), with potential organic inclusions and clay minerals that may contribute to the soil exchange complex in the active
      layer, and reservoirs of exchangeable cations with readily available nutrients (Ca2+, K+, Mg2+) for plant uptake. Overall,
      changes in the active layer thickness (ALT) and in the balance between organic and mineral constituents of the active layer

are likely to influence the size of the reservoir and the sources of available nutrients for plant uptake, and thereby promote the
      tundra vegetation productivity. However, these changes remain poorly quantified and further estimation of the exchangeable
      nutrient (Ca2+, K+, Mg2+) reservoir about to thaw relative to the current stock in the active layer is a needed step for ecosystem
      models simulating the evolution of vegetation development and microbial activity upon permafrost thaw (Fisher et al., 2014;
      Koyama et al., 2014; Sulman et al., 2021; van der Kolk et al., 2016).


      The main objective of this study is to investigate the influence of permafrost thaw on the properties of the soil exchange
      complex. We posit that permafrost thaw exposes a deep soil layer with contrasted properties of the soil exchange complex
      relative to the seasonally thawing active layer, and we aim to quantify this contrast. Across a range of permafrost soil profiles
      with contrasted active layer thickness, we investigate the difference in the constituents controlling the soil capacity to retain

exchangeable cations and the distribution of exchangeable cations retained, and we quantify the stocks in exchangeable cations
      in the seasonally thawed active layer and in the permafrost.

## 2. Material and methods

### 2.1 Study area and sampling

The study is conducted within the Eight Mile Lake (EML) watershed close to Healy, in Interior Alaska, USA (63°52'42N,

149°15'12W; Schuur et al., 2009). The research site is underlain by degrading permafrost in the discontinuous permafrost zone
      (Natali et al., 2011; Osterkamp et al., 2009) and covers a natural gradient in permafrost thaw (Gradient site). Long-term (1977-
      2015) mean annual air temperatures range between 10.2 ± 3.8 °C (for the growing season) and -16.6 ± 2.5°C (for the non-
      growing season) and the mean annual precipitation is about 381 mm (Natali et al., 2011; Vogel et al., 2009). Soils at EML are
      classified as Turbic Histic Cryosols (Mulligan, 2017) and characterized by a 35 to 55 cm thick organic layer (SOC ≥ 20%) in

surface. This organic horizon overlays a cryoturbated mineral soil (SOC < 20%) composed of glacial till and loess parent
      material (Hicks Pries et al., 2012; Osterkamp et al., 2009; Vogel et al., 2009) with dominant amounts of quartz and feldspars
      (Plaza et al., 2019). The site is located on moist acidic tundra, with a dominance of sedges (as *Eriophorum vaginatum* L. and
      *Carex bigelowii* Torr. ex Schwein), evergreen shrubs (e.g., *Andromeda polifolia* L., *Rhododendron tomentosum* Harmaja,
      *Vaccinium vitis-idaea* L., and *Empetrum nigrum* L.), deciduous shrubs (e.g., *Vaccinium uliginosum* L. and *Betula nana* L.)

and forbs (e.g., *Rubus chamaemorus* L.). Non-vascular plant cover is dominated by mosses (mainly *Sphagnum* spp., *Dicranum*
      spp., and feather mosses including *Hylocomium splendens* and *Pleurozium schreberi*) and lichen species (e.g., *Nephroma* spp.,

*Cladonia* spp., and *Flavocetraria cucullata*) (Deane-Coe et al., 2015; Natali et al., 2012; Schuur et al., 2007). Since the start of the thermokarst development, vegetation cover changed with the evergreen and deciduous shrubs (as *V. uliginosum* and *R. tomentosum*), and forbs (as *R. chamaemorus*) being dominant at the expense of the sedges (Jasinski et al., 2018; Mauclet et al., 2021; Schuur et al., 2007; Villani et al., 2022).

A field campaign took place at EML at the late season period between mid-August and early-September in 2019 to sample permafrost soil profiles with contrasts in their maximal thaw depth. Seven soil cores were collected and their respective active layer thicknesses (ALT) were measured with a metal probe (Table S1). Soils were sampled to a maximum depth of 120 cm and subdivided into 5 cm to 10 cm horizons. Active layer samples were collected using a hammer and chisel as deep as possible (up to 45 cm, unless water table was higher than 45 cm). Below 45 cm depth, active layer and permafrost samples were collected using a steel pipe (diameter 4.5 cm) that was manually hammered into the soil using a sledgehammer (Palmtag et al., 2015). The demarcation between organic and mineral horizons was determined visually and confirmed by %OC analysis (section 2.2) when the %OC of the soil decreased to less than 20% (Hicks Pries et al., 2012). For each soil core, we kept separated samples from organic active layers, mineral active layers, and permafrost soil layers. In the lab, soil samples (n=85) were dried at air temperature in a ventilated and temperature-controlled room. Mineral horizons (SOC ≤ 20%) were sieved at 2 mm and organic horizons (SOC > 20%) free of large roots were ground.

## 2.2 Characterization of the total soil fraction

### 2.2.1 Soil pH

The soil $pH_{H2O}$ and $pH_{KCl}$ were measured on all soil samples (n=85) with the pH probe (Inlab micro) connected to the pH-meter (Mettier Toledo SevenCompact DuoS213). Mineral soil samples were mixed in the 1:5 proportion with $H_2O$ or 1M KCl (Peech, 1965), whereas organic soil samples required adaptation and were mixed in 1:15 proportion with $H_2O$ or 1M KCl in order to have enough liquid solution for the insertion of the pH probe. The pH probe was calibrated for pH 4 and 7 before measurement.

### 2.2.2 Soil organic carbon content

The total soil carbon content was measured on all soil samples (n=85) with the C, N, S elemental analyzer vario EL CUBE (ELEMENTAR ®, Germany). The C measurements on soil samples (n=85) are reported to the dry soil matrix (105°C). Each sample was measured twice and the average standard deviation for C content ~5%, with the detection limit <0.1%. As the presence of carbonates was not detected by X-ray diffraction, the total soil carbon content is considered equivalent to soil organic carbon (SOC) content. The SOC content analysis was used to confirm the limit between organic (SOC > 20%) and mineral (SOC ≤ 20%) soil horizons (Hicks Pries et al., 2012).



Previous studies at the EML research site (Interior Alaska, USA) published datasets on SOC content ($g_C$ $kg_{soil}^{-1}$) and soil bulk

densities (BD, in $g_{soil}$ $cm^{-3}$) on permafrost soil profiles collected between 2009 and 2013 (Plaza et al., 2017). Because of the

strong correlation between the soil organic matter content and the soil bulk density (Askin and Özdemir, 2003; Chaudhari et

al., 2013; Périé and Ouimet, 2008), we established a linear regression (Eq. 1; R² = 0.73) between published SOC content and

BD measurements on soil samples at EML (including active layer and permafrost layers; Plaza et al., 2017). We applied the

resulting empirical equation of correlation between the two parameters (Eq. 1) to our newly available data of SOC for soil

samples collected at the Gradient site in 2019.

$BD_i = 0.92 - 0.0018 \times SOC\ content_i$                                                     (1)

With $i$ the sample horizon considered, and $SOC\ content$ in ($g_C$ $kg_{soil}^{-1}$).

Data of the obtained bulk density ($BD$, in $g_{soil}$ $cm^{-3}$) are presented in the Table S2. From the SOC content and the estimated

BD, we calculated the stocks in SOC ($kg_C$ $m^{-2}$) within the first meter of our soils with the equation Eq. 2:

$Stock_{SOC} = \sum_{i=0\ cm}^{i=100\ cm} \left[ SOC\ content_i \times BD_i \times thickness_i \times \frac{1}{100} \right]$                     (2)

With $i$ the sample horizon considered, $SOC\ content$ in $g_C$ $kg_{soil}^{-1}$, $BD$ in $g_{soil}$ $cm^{-3}$, and $thickness$ in cm.

To evaluate the total SOC stock within the first meter of incomplete soil profiles (as Min1, Min3, Mod1), BD and SOC content

of the missing soil horizons were approximated as the average value of the two adjacent lower and upper horizons (Table S2).

**2.2.3 Soil elemental composition**

On three selected soil profiles covering contrasted ALT (Min1, Mod3, Ext3), the total concentration in Ca, K, Mg, and Na in

bulk soils was determined by inductively coupled plasma optical emission spectroscopy (ICP-OES, iCAP 6500 ThermoFisher

Scientific, Waltham, USA) after alkaline fusion (Chao and Sanzolone, 1992). The accuracy on mineral element (Ca, K, Mg,

and Na) analyses was assessed using trueness ($\pm$ 2%, $\pm$ 5%, $\pm$ 2%, and $\pm$ 2%; respectively) on the USGS basalt reference

material BHVO-2 (Wilson, 1997) and the analytical precision ($\pm$ 0.5%) for each element. The limits of detection (LOD) were

0.05 mg L$^{-1}$, 0.01 mg L$^{-1}$, 0.001 mg L$^{-1}$, and 0.02 mg L$^{-1}$ for Ca, K, Mg, and Na, respectively. The blank levels were below the

detection limit for Ca, K, Mg, Na. The sum the total concentration in Ca, K, Mg, and Na was calculated as the total reserve in

bases (TRB expressed in cmol$_c$ kg$^{-1}$; Herbillon, 1986).

For two key plant nutrients, Ca and K, the total concentration in soils were determined in all soil profiles using a non-destructive

portable X-ray fluorescence (pXRF) device Niton$^{TM}$ XL3t GOLDD+ (Thermo Fisher Scientific, Waltham, USA). The pXRF

analyses were conducted in laboratory conditions, using a lead stand to protect the operator from X-rays emission. For the

measurement, the dried sample powder was placed on a circular plastic cap (2.5 diameter) provided with a transparent thin

film (prolene 4μm) at its base, in order to reach ~1 cm of sample thickness. Each sample was scanned for a total measurement



time of 90 seconds and the precision of the method was evaluated by five repeated measurements on 31 soil samples representative of the organic and mineral layers. The relative standard deviation (i.e., standard deviation divided by the mean, expressed in % to the mean) was ~2% for Ca and K concentration. The trueness of the pXRF measurements was obtained after data correction using a calibration model between pXRF and accurate ICP-OES measurements (Monhonval et al., 2021b).

### 2.2.4 Soil mineralogy

We assessed the mineralogy of 21 bulk samples from five selected ~1m deep permafrost soil profiles that cover contrasted active layer thicknesses across the Gradient site (Min3, Mod2, Mod3, Ext2, Ext3). The mineralogy of bulk samples was determined by X-ray diffraction on finely ground powder (Bruker D8 Advance, Cu Kα, 40 kV, 30 mA, 2θ = 1/min, detection limit of 5%).

### 2.3 Characterization of the exchangeable soil fraction

### 2.3.1 Soil cationic exchange capacity, base saturation and exchange acidity

The concentration in exchangeable base cations and the potential CEC of soil were determined on all soil samples (n=85). The exchangeable base cations ($Ca^{2+}$, $K^+$, $Mg^{2+}$, $Na^+$) were collected at neutral pH by ammonium acetate extraction (1M $NH_4OAc$, Metson, 1956) and their concentration within the extracts was determined by ICP-OES. The potential soil CEC (expressed in reference to the dry weight at 105°) was determined by desorbing the ammonium from the soil exchange complex and

measuring the amount of ammonium recovered by spectrophotometry (Spectroquant® ammonium test kits). The base saturation (BS; percentage of CEC occupied by exchangeable base cations $Ca^{2+}$, $K^+$, $Mg^{2+}$, and $Na^+$) was calculated with the equation Eq. 3:

$$BS = \frac{\sum(Ca^{2+} + K^+ + Mg^{2+} + Na^+)}{CEC} \qquad (3)$$

For two selected soil profiles with contrasted ALT (n=26; Min3 and Ext3), the effective CEC at the soil pH and the exchange acidity ($Al^{3+}$ and $H^+$) were determined through $BaCl_2$ extraction, following the method from Hendershot and Duquette (1986).

### 2.3.2 Estimates for CEC density and stocks in total and exchangeable elements

While the usual soil CEC ($cmol_c$ $kg^{-1}$) is relative to soil mass, we evaluated here the CEC density relative the soil volume (in $cmol_c$ $m^{-3}$) along permafrost soil profiles. For the calculation, we relied on the soil CEC (in $cmol_c$ $kg^{-1}$) and bulk density (BD

in g $cm^{-3}$) with the equation Eq. 4:

$$CEC\ density_i = CEC_i \times BD_i \times 1000 \qquad (4)$$

With $i$ the sample horizon considered.





We also evaluated the stock (in g m$^{-3}$) of total elements (Ca, K, Mg, and Na) and exchangeable base cations (Ca$^{2+}$, K$^+$, Mg$^{2+}$,

and Na$^+$) within soils using the mineral element concentration of soil [X] (g kg$^{-1}$) and the bulk density (BD in g cm$^{-3}$) with the

equation Eq. 5:

$$stock\ X_i = [X]_i \times BD_i \times 1000 \qquad (5)$$

With $i$ the sample horizon considered.

## 3.  Results and discussion

**3.1 Distribution of the organic and mineral constituents from the soil exchange complex within permafrost soil profiles**

The soils from EML accumulate organic matter, and show contrasts in SOC contents between the organic surface (SOC >

20%) and deep mineral (SOC ≤ 20%) soil horizons (Fig. 1). This is consistent with expectations for typical sub-Arctic tundra

soils (Christensen et al., 1999; Hugelius et al., 2014; Michaelson et al., 1996). On average, organic soil horizons hold about

43.5% OC, whereas mineral soil horizons hold about 4.5% OC (Table S2). For some soil profiles, increase in SOC content at

depth (up to 30% OC) may reflect the presence of old organic matter inclusions within the soil mineral phase (Fig. 1).

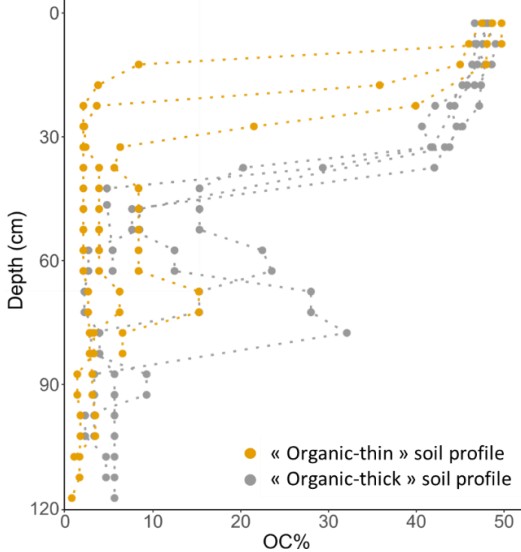

**Figure 1: Distribution of soil organic carbon content (%OC) into permafrost soil. The "organic-thin" (in orange) soil profiles present
more shallowly distributed high values for OC content, and the "organic-thick" (in grey) soil profiles present more deeply distributed**
**high values for OC content.**

Results suggest a difference in the SOC accumulation and distribution within soil profiles related to the active layer thickness

(Fig. 2): soil profiles with shallower active layer (ALT ≤ 60 cm) have a higher organic layer thickness (OLT ≥ 40 cm) whereas

soil profiles with deeper active layer (ALT > 60 cm) have a thinner organic layer in surface (OLT < 40 cm). This can be

explained by the low thermal diffusivity and conductivity of the organic matter (Adams, 1973; Farouki, 1981) that insulates





the soil (Decharme et al., 2016; Lawrence and Slater, 2008) and thereby influences the permafrost thaw depth. As a result, the "organic-thick" soil profiles are characterized by a shallower active layer (ALT ≤ 60 cm; Min1, Min3, Mod1, Ext1) than the "organic-thin" soil profiles with a deeper active layer (ALT > 60 cm; Min2, Mod2, Mod3, Ext2, Ext3) (as illustrated by Fig. 3). The difference in patterns of SOC distribution between the "organic-thick" and "organic-thin" permafrost soil profiles suggests a potential loss in C with permafrost degradation, as reported in other Arctic and sub-Arctic sites (Pegoraro et al.,

2021; Plaza et al., 2019; Schuur et al., 2021). This loss can tentatively be attributed to (i) an increase in organic matter decomposition and carbon emission (concomitant to microbial activity stimulation; Aerts, 2006; Hicks Pries et al., 2012; Mack et al., 2004; Nadelhoffer et al., 1992; Schuur et al., 2021), and (ii) the increase in soil drainage capacity upon permafrost thaw (Gebauer et al., 1996; Ping et al., 2015, 1998) that may contribute to carbon leaching and lateral transport from the thawed soil horizons (Plaza et al., 2019).


Across the site, the stock of OC within the first meter of permafrost soil average ~45 $kg_C \, m^{-2}$ (Table S2) which is comparable to previous stock evaluation at the same study site (EML, Interior Alaska, USA) in 2004 (over 50kg C m$^{-2}$; Hicks Pries et al., 2012). More specifically, the "organic-thick" soil profiles accumulate around 45-65 $kg_C \, m^{-2}$ in the top soil meter, and the "organic-thin" soil profiles accumulate between 22 and 33 $kg_C \, m^{-2}$ (with one exception for Mod1 that reaches 60 $kg_C \, m^{-2}$;

Table S2).

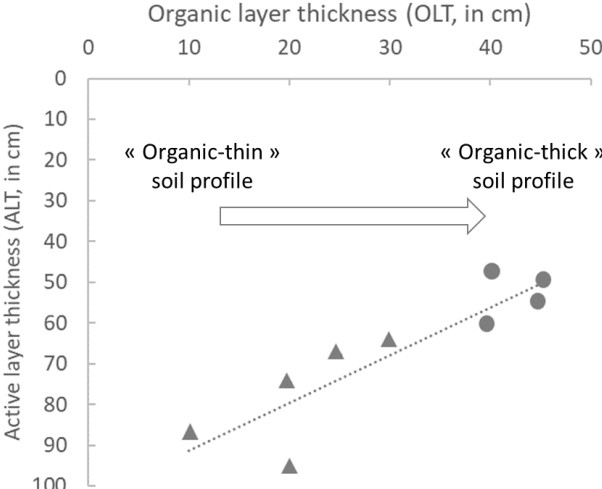

**Figure 2: Relationship between organic layer thickness (OLT) and active layer thickness (ALT) of permafrost soil profiles along a thermokarst gradient at Eight Mile Lake (Interior Alaska, USA) with "organic-thick" (OLT ≥ 40 cm; circle) and "organic-thin"**
**(OLT < 40 cm; triangle) soil profiles.**

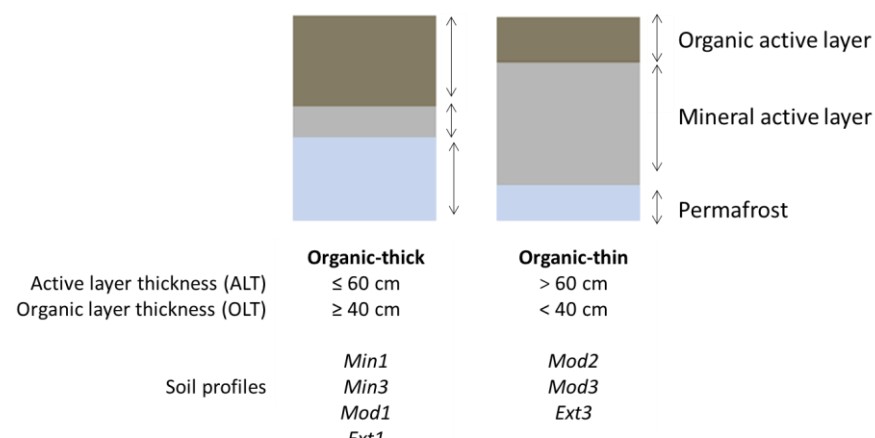

**Figure 3: Schematic representation of the contrasted permafrost soil profiles at Eight Mile Lake (Interior Alaska, USA), with organic-thick soil profiles (OLT ≥ 40 cm and ALT ≤ 60 cm) and organic-thin soil profiles (OLT < 40 cm and ALT > 60 cm).**

In parallel to the decrease in OC content with depth, the analysis of the soil composition in mineral elements reveals an increase
(3- to 8-times) in the Ca, K, Mg, and Na total concentrations between the organic-rich surface horizons and the deeper mineral soil horizons at the Gradient site (as illustrated for one soil profile in Fig. 4). On average, soil surface horizons contain ~2.5 g kg$^{-1}$ Ca, ~1.4 g kg$^{-1}$ Mg, ~4.4 g kg$^{-1}$ K, and ~0.9 g kg$^{-1}$ Na, and deep horizons contain about ~8 g kg$^{-1}$ Ca, ~7.2 g kg$^{-1}$ Mg, ~16 g kg$^{-1}$ K, and ~11.2 g kg$^{-1}$ Na (Table S3).

This net difference in the total concentrations in Ca, K, Mg and Na between surface and deep soil horizons reflects the transition between organic and mineral soil horizons (at about 40 cm depth for Min1; Fig. 4), with higher elemental concentrations in the mineral soil horizons originating from the mineral constituents. In particular, the mineral phases observed at EML include primary minerals (i.e., quartz, micas, feldspar-K, plagioclase, and amphibole) and secondary minerals (i.e., kaolinite, vermiculite, and illite) (Table S4). While some secondary minerals (as clay minerals, Al and Fe oxides) may directly contribute
to the soil exchange complex, primary minerals constitute a soil reserve in potentially weatherable minerals that may release base cations into the soil solution. Therefore, the soil composition and distribution in minerals within permafrost will directly influence the soil exchange properties upon active layer thickening.

Overall, our data support the hypothesis of changing balance between organic and mineral constituents forming the soil
exchange complex upon permafrost degradation. A thinner organic layer in highly thawed permafrost soils leads to a lower contribution from organic constituents to the soil exchange complex in the active layer of these more degraded soils relative to poorly thawed soils. In addition, permafrost thaw exposes mineral layers containing proportionally less organic matter and more clay minerals than the above active layer, leading to an increasing contribution from clay minerals to the soil exchange complex at increasing depth, and providing new sites for cation exchange.

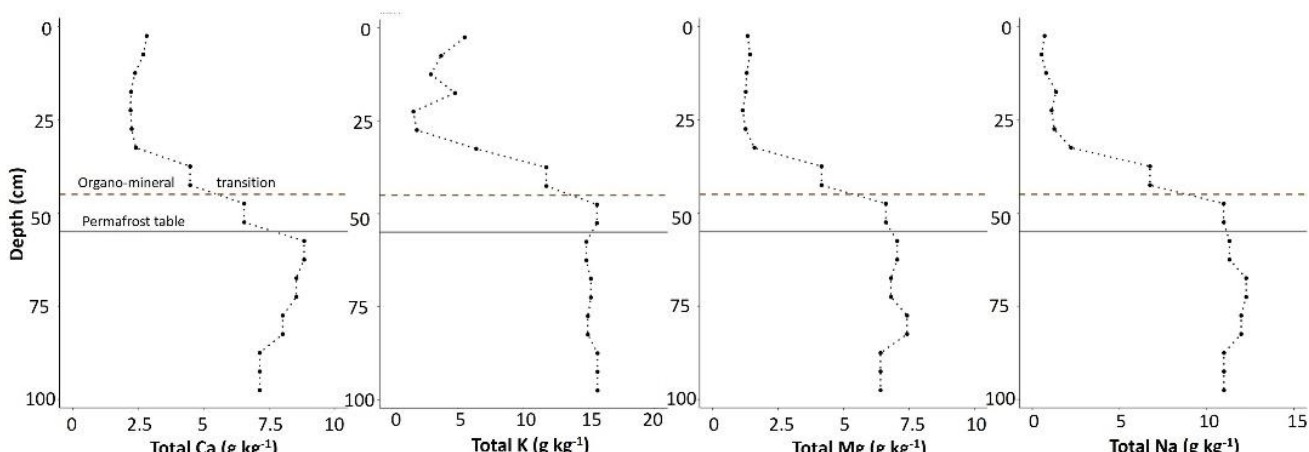


**Figure 4: Distribution of concentrations in total Ca, K, Mg, and Na (g kg$^{-1}$) in an "organic-thick" permafrost soil profile (Min1). Permafrost table is indicated by the straight grey line, and the transition between the organic (SOC ≥ 20%) and mineral (SOC < 20%) soil is indicated by the dotted brown line.**

**3.2 Variation of the soil exchange properties along permafrost soil profiles**

**3.2.1 Change in CEC distribution and CEC density with soil constituents**

Given that the distribution of soil CEC follows the SOC content distribution ($R^2$ = 0.91), high CEC values are more deeply distributed in the "organic-thick" (OLT ≥ 40 cm) than in the "organic-thin" (OLT < 40 cm) permafrost soil profiles (Fig. 5a). Overall, our CEC values measured in organic horizons (~90 ± 24 cmol$_c$ kg$^{-1}$) and mineral horizons (~20 ± 6 cmol$_c$ kg$^{-1}$; data from Table S3) are in the same range than the values reported in arctic tundra soils underlain by permafrost (Ping et al., 2005,

1998; CEC ~88 ± 36 cmol$_c$ kg$^{-1}$ for organic soil layers and ~24 ± 12 cmol$_c$ kg$^{-1}$ for mineral soil layers). Locally, we observe high CEC values (~45-50 cmol$_c$ kg$^{-1}$) at 60-80 cm depth in two soil profiles (Min3 and Mod1; Fig. 5a), and this reflects deep organic inclusions (Fig. 1). Below the organic-mineral transition, given that the concentration in organic carbon decreases and the presence of clay minerals is verified (vermiculite, kaolinite, illite; Table S4), the contribution from the clay minerals to the CEC increases.


Furthermore, soil CEC is dictated by the presence of variable charges provided by organic matter, some clay minerals and Fe and Al oxides. While the organic matter is usually associated with a variable charge, clay minerals are assumed to have both constant and variable charges (Kamprath and Smyth, 2005; Weil and Brady, 2016). As the strength of the variable charge depends on ionic strength and pH, chemical soil environment directly influences the soil variable charge and thereby the soil

CEC. In our soils, the average pH$_{H2O}$ oscillates between ~4.0 in surface and ~5.3 at depth (Table S2). The low pH in the organic-rich soil surface (pH$_{H2O}$ ~ 4) is expected to reduce the number of negatively charged exchange sites on the variable charge components (i.e., organic matter and clay minerals as kaolinite) and thereby reduce the effective CEC compared to the deep mineral horizons with higher pH (pH$_{H2O}$ ~5.3) and less variable charge components (Bigorre et al., 2000).





However, we should note that the method used for the potential CEC measurement at neutral pH (method of Metson, 1956) does not consider the soil pH effect on CEC and may thereby overestimate the effective CEC of the soil. In particular, we observe peaks in potential CEC at about 25 cm depth in three soil profiles (Min1, Min3, Mod1; Fig. 5a) that seem to be correlated with the specific water table depth and accumulation of Fe-oxides (Fig. S1; data from Monhonval A.). The accumulation of Fe-oxides at the level of the water table may be explained by the translocation of reduced Fe upon water

saturated condition and Fe oxide precipitation in favorable oxic conditions (Herndon et al., 2017; Monhonval et al., 2021a). However, the acidic soil conditions at that depth ($pH_{H2O}$ ~ 4) protonate Fe oxides and prevent them from contributing to the effective soil CEC. This is further highlighted by the measurements of effective CEC on two selected profiles (Min3 and Ext3; Fig. S2a): the effective CEC is consistently lower than the potential CEC and the difference is much pronounced in surface than at depth. More specifically, the effective CEC is between 50 and 90% lower than the potential CEC in the organic horizons

where the soil $pH_{H2O}$ is at ~ 4, and between 20 and 60% different in the mineral permafrost where the soil $pH_{H2O}$ is at ~5.3.

Unlike the distribution of soil CEC values within soil profile, the estimates for CEC density show lower values in the organic active layers (~9 400 ± 6 300 $cmol_c$ $m^{-3}$) than in the mineral soil horizons (~16 000 ± 4 000 $cmol_c$ $m^{-3}$ for mineral active layer; ~12 000 ± 3 500 $cmol_c$ $m^{-3}$ for permafrost) (Table S5). More specifically, estimates for CEC density increase along the organic

active layer and stabilize in the mineral soil horizons (Fig. 5b). When we integrate the CEC density over the permafrost soil profile top meter, "organic-thick" soil profiles have lower CEC density (~12 000 $cmol_c$ $m^{-2}$) than the "organic-thin" soil profiles (~14 000 $cmol_c$ $m^{-2}$). This is mainly explained by the difference in organic matter contribution to the total CEC density: the uppermost 40 cm of "organic-thick" soil profiles have lower CEC density (~3 500 $cmol_c$ $m^{-2}$) than the "organic-thin" soil profiles (~5 000 $cmol_c$ $m^{-2}$).


The major change in the estimates for CEC density between soil surface and depth is driven by the massive increase in bulk density between organic and mineral horizons. The increase in soil bulk density with depth overcomes the lower CEC values, and results in higher estimates for CEC density. Within the top soil meter, the thin organic-rich surface layer (OLT < 40 cm) generates a higher contribution of clay mineral to the CEC density than the thicker organic matter accumulation (OLT ≥ 40

cm). Overall, our results support that SOM accumulation in surface influences both, the ALT (Fig. 2) and the distribution of CEC density (Fig. 5b), in permafrost soil profiles. Therefore, the potential decrease in OLT upon warming may increase the average CEC density of the active layer by increasing the relative clay mineral contribution to the CEC. As a result, we expect the potential increase in CEC density with permafrost thaw to increase the potential for plant nutrient retention within the active layer.


Earth System
Science
Data

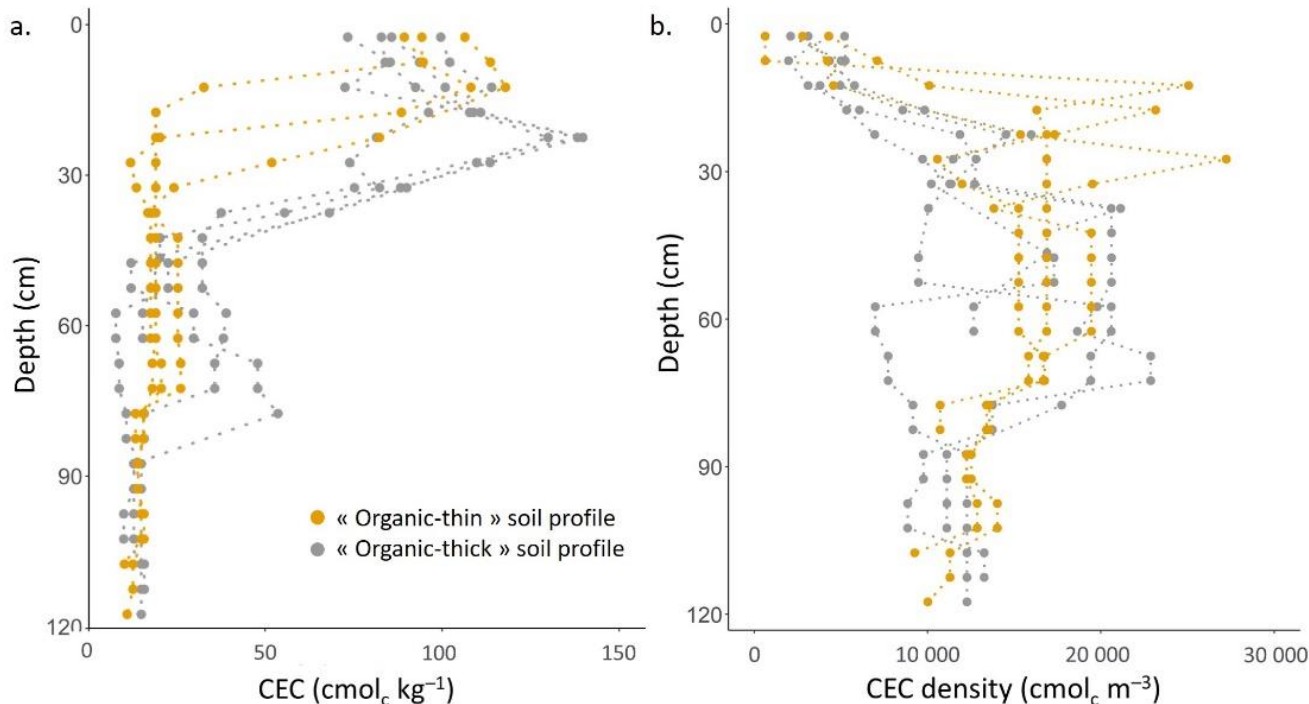

**Figure 5: (a) Distribution of the soil cation exchange capacity (CEC, in cmol$_c$ kg$^{-1}$) and (b) distribution of the density of soil cation exchange capacity (CEC density, in cmol$_c$ m$^{-3}$) with depth between "organic-thin" (in orange, n=34) and "organic-thick" (in grey, n=51) permafrost soil profiles. Estimation of soil CEC density is the product between soil bulk density and CEC.**

### 315  3.2.2 Influence of soil pH and total reserve in bases (TRB) on the base saturation

Within permafrost soils at EML, we observe an increase in the base saturation (BS) of the soil exchange complex from the organic active layer to the deeper mineral soil horizons (Fig. 6a). On average, active layers show base saturation around ~20% in the organic soil surface and ~35% in the underlying mineral horizons, while permafrost soil horizons show base saturation around ~70% (Table S3).


The overall increase in BS with depth in our soils suggests an increase in the base cation availability between active layer and permafrost. These results are consistent with another study showing larger concentration of exchangeable K and Ca within permafrost than active layer soils (Keller et al., 2007) due to cation leaching with seasonal thaw of the active layer (Frey and McClelland, 2009; Walvoord and Kurylyk, 2016). More specifically, BS distribution within permafrost soil layers reveals a
lower BS in upper permafrost soil (i.e., permafrost soil horizons between 0 and 20 cm below the permafrost table) than in the deep permafrost soil (i.e., permafrost soil horizons at least 20 cm below the permafrost table) (Fig. 6a). This points that previous warming events may have thawed surface layers of the current permafrost in the so called "transient layer", i.e., the zone between the base of the active layer and long-term permafrost where thaw is less frequent and occurs in response to a climate





shift, disturbance of the organic layer or change in surface vegetation (Shur et al., 2005). These rare thaw events have likely
favoured the leaching of the soil base cations of this layer (Lamhonwah et al., 2017).

The distribution of BS along permafrost soil profiles shows a positive linear relation with soil pH ($R^2$ = 0.67; Fig. 6b), well in line with previous studies across various ecosystems (Beery and Wilding, 1971; Binkley et al., 1989; Giesler et al., 1998; Thomas, 2019) including the Arctic (Ping et al., 2005). This reflects the balance between acid cations and base cations on the
soil exchange complex. As the BS contributes to the soil buffer capacity (Bowman et al., 2008; Tian and Niu, 2015; Ulrich, 1983), the more acidic conditions in permafrost soil surface ($pH_{H2O}$ ~ 4.0) reveal lower soil buffer capacity and thereby lower BS (green dots on Fig. 7b). In contrast, less acidic soil conditions at depth ($pH_{H2O}$ ~5.3) reflect higher soil buffer capacity and thereby higher BS (grey dots on Fig. 6b). This is further supported by the exchange acidity measured for two soil profiles (Min3 and Ext3) that shows contrasted values along the profiles with higher concentrations in exchangeable acid cations ($Al^{3+}$
and $H^+$) in surface than at depth (Fig. S2b). This supports the general assumption that major exchangeable cations are $Al^{3+}$, $H^+$, $Ca^{2+}$, $Mg^{2+}$ and $K^+$ in the more acidic soil surface, whereas major exchangeable cations are $Ca^{2+}$, $Mg^{2+}$, $K^+$, and $Na^+$ in the less acidic soils (Havlin, 2005). The lower values of $pH_{KCl}$ (between 3.29 in surface and 4.47 at depth) than $pH_{H2O}$ (between ~4.02 in surface and ~5.27 at depth; Table S2) also support the presence of exchangeable acid cations adsorbed onto the soil exchange complex.


The difference in BS between the organic soil surface and the deep permafrost soil likely reflects the increase in the weatherable mineral reserve in the soil with depth, quantified by the total reserve in bases (TRB; Herbillon, 1986). The net increase in TRB with depth (higher values in the deep permafrost soil ~190 $cmol_c$ $kg^{-1}$ than in the organic soil surface ~45 $cmol_c$ $kg^{-1}$; Table S3) follows the organo-mineral transition and reflects higher soil reservoirs of total Ca, K, Mg, and Na in the mineral soil
layers compared to the organic active layer. In particular, BS in the mineral soil horizons increases gradually between the mineral active layer, the upper permafrost horizons and the deep permafrost horizons (Fig. 6a) whereas the TRB remains constant (Fig. 6c). This likely reflects the influence of permafrost thaw on the leaching of exchangeable base cations ($Ca^{2+}$, $K^+$, $Mg^{2+}$, and $Na^+$) that are soluble cations. The seasonally thawed mineral active layer is subject to more cation leaching than the upper permafrost (potentially thawed upon exceptional previous warming), and more than the deep permafrost.


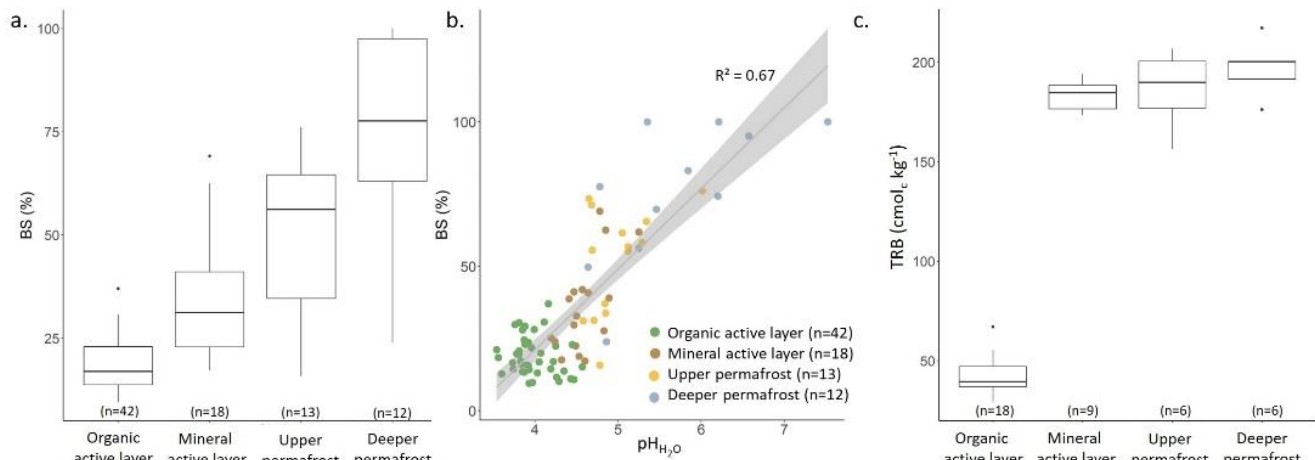

**Figure 6: (a) Distribution of base saturation (BS, in %), (b) linear correlation between BS and soil pH, and (c) distribution of the total reserve in base (TRB in cmol$_c$ kg$^{-1}$; Herbillon, 1986) for the soil samples collected at the Gradient site (Eight Mile Lake, Interior Alaska, USA) in 2019. The upper permafrost consists of soil samples less than 20 cm below the permafrost and the deeper permafrost consists of soil samples at least 20 cm below the permafrost table.**

### 3.2.3 Change in the distribution of exchangeable base cations

The fractions of exchangeable base cations (considered as readily available for plant uptake) over the total concentration of these elements in soils are between 4 and more than 15 times higher in organic than mineral soil layers, depending on the element (Ca, K, Mg, and Na). On average, from the total Ca, K, Mg, Na concentrations (n=7 for Ca and K, n=3 for Mg and Na), we observe that ~75%, ~15%, ~45% and ~25% is exchangeable in organic soil layers, and ~15%, <1%, ~4% and ~2% is exchangeable in mineral soil layers, respectively (Table S5). These results are well in line with the proportions of exchangeable base cations reported for an Alaskan tundra site, such as >70% for Ca$_{exch}$ and 18% of K$_{exch}$ in organic soil horizons, and ~23% for Ca$_{exch}$ and 1% for K$_{exch}$ in mineral soil horizons (Chapin et al., 1979).

To investigate the distribution of stocks in exchangeable base cations within soil profiles, we converted the concentrations in exchangeable base cations into stocks (in g m$^{-3}$). At EML, permafrost soil profiles show an increase in stock of exchangeable base cations (Ca$^{2+}$, K$^+$, Mg$^{2+}$ and Na$^+$ in g m$^{-3}$) with depth (Fig. 7a-c) that follows the increase in BS (Fig. 7b-d). In particular, stocks of each exchangeable base cation increase from organic active layer to deep mineral permafrost as: from 190 to 920 g m$^{-3}$ for Ca$^{2+}$; from 40 to 50 g m$^{-3}$ for K$^+$; from 50 to 250 g m$^{-3}$ for Mg$^{2+}$; and from 30 to 160 g m$^{-3}$ for Na$^+$ (Table S5). Furthermore, we observe a difference in the distribution of base cation stocks between the "organic-thick" permafrost soils showing low stocks in base cations more deeply distributed (≥ 40 cm deep) than the "organic-thin" permafrost soils (< 40 cm deep) (as illustrated in Fig. 7a-c, respectively).



Overall, the distribution of base cation stocks follows the soil distribution of CEC density and BS, and results in much higher (between 1.5 to more than 4-times) reserves in exchangeable base cations ($\Sigma$ Ca$^{2+}$+K$^+$+Mg$^{2+}$+Na$^+$) at depth than in surface. Upon warming, permafrost thaw exposes deeper permafrost soil horizons to above zero temperatures and thereby increases the potential soil capacity to retain base cations readily available for plant uptake (i.e., soil CEC density) and provides newly thawed pool of nutrient base cations. As Arctic ecosystems only rely on active layer thickness in providing essential nutrients

to sustain vegetation growth and development (Iversen et al., 2015; Ping et al., 1998), the contrast in nutrient base cation stocks between the organic and the mineral parts of the active layer is of great importance for changes in tundra vegetation productivity.

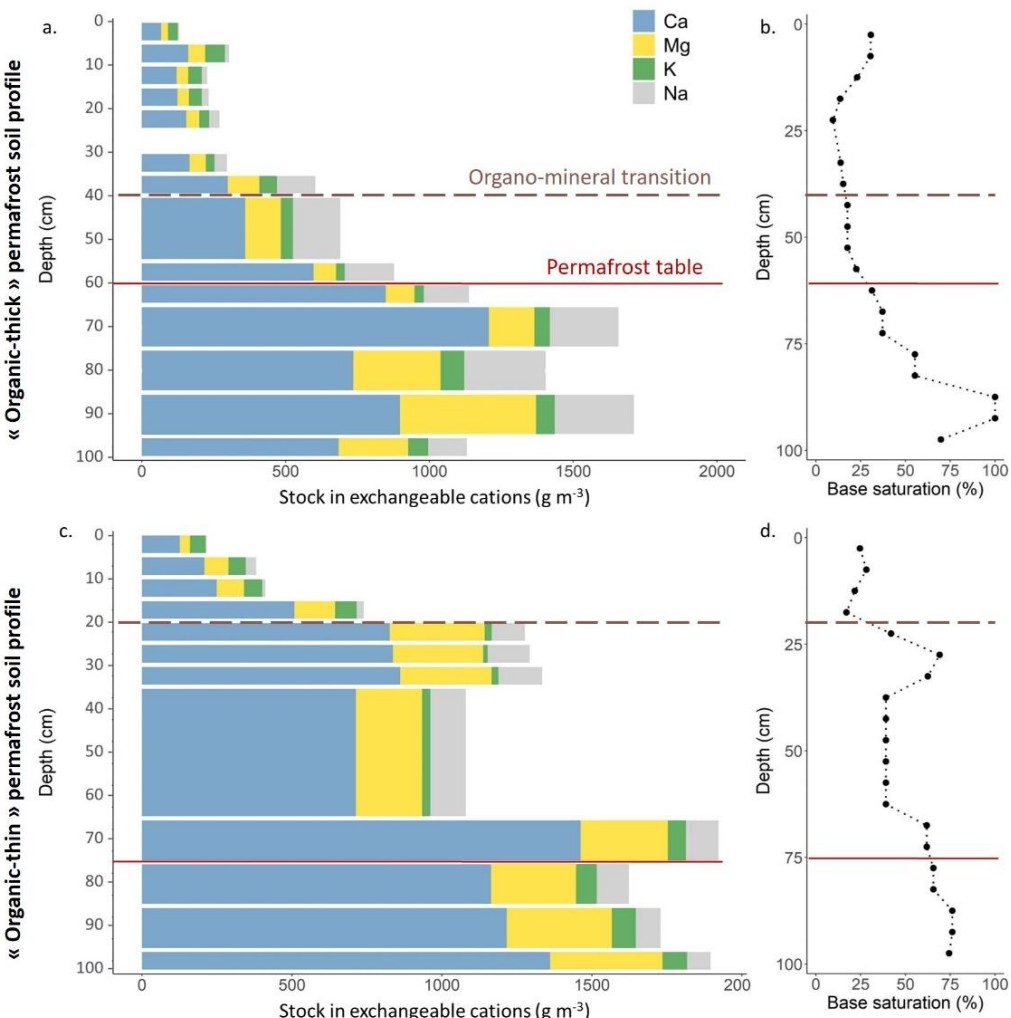

**Figure 7: (a-c) Distribution of stocks in exchangeable base cations (Ca$^{2+}$, Mg$^{2+}$, K$^+$, Na$^+$, in g m$^{-3}$), and (b-d) distribution of base**
**saturation (%) along "organic thick" (Min3; n=14) and "organic-thin" (Mod3; n=12) permafrost soil profiles at the Gradient site (Eight Mile Lake, Interior Alaska, USA). Red lines indicate the permafrost table and brown dotted lines indicate the transition between organic and mineral soil horizons.**





**3.3 Influence of vegetation nutrient cycling on the stocks in exchangeable base cations within permafrost soil profiles**

In this section, we investigate the ecological influence of vegetation nutrient uptake and cycling on the distribution of specific exchangeable base cation stocks (in g m$^{-3}$) within permafrost soil profiles. When looking at the stocks of the four exchangeable base cations individually, we observe different trends in their vertical distributions into active layer soil horizons. Stocks in exchangeable Ca, Mg, and Na are low in the organic part of the active layer, and increase with depth. On average, these stocks are about ~3 (for $Ca_{exch}$ and $Mg_{exch}$) and ~7 (for $Na_{exch}$) times lower in organic soil layers than in mineral active layer horizons

(as illustrated in Fig. 8a-b-e-f-g-h). Conversely, stocks in $K_{exch}$ are ~1.5-times higher in the organic surface horizons than in mineral soil horizons (as illustrated in Fig. 8c-d). As the exchangeable base cation stocks follow the transition between organic and mineral soil horizons, the low (for $Ca_{exch}$, $Mg_{exch}$, $Na_{exch}$) and high (for $K_{exch}$) values of stock are more deeply distributed in "organic-thick" than in "organic-thin" permafrost soils.

While the distributions in $Ca_{exch}$, $Mg_{exch}$, and $Na_{exch}$ stocks with depth reflect a depletion in nutrient base cations within the organic soil surface relative to the mineral soil horizon, the singular vertical distribution of $K_{exch}$ across the active layer suggests a nutrient uplift through plant nutrient uptake and cycling, known as biolifting process (Jobbágy and Jackson, 2001). Our results are consistent with the vertical distribution reported in the literature for available nutrients from shallowest to deepest soil horizons in the following order $K_{exch} > Ca_{exch} > Mg_{exch} > Na_{exch}$ across a wide variety of ecological conditions (Jobbágy

and Jackson, 2001). This supports the key role of plant cycling on the vertical distribution of plant limiting nutrients such as K, and thereby on the vegetation production and development within nutrient-limited ecosystems (Flanagan and Cleve, 1983; Hobbie, 1992; Hobbie et al., 2002; Nadelhoffer et al., 1992; Poszwa et al., 2000).





**Figure 8: Vertical distribution of stocks in exchangeable base cations (Ca²⁺, K⁺, Mg²⁺, and Na⁺ in g m⁻³) in "organic-thick" (Min3; n=14) and "organic-thin" (Mod3; n=12) permafrost soil profiles at the Gradient site (Eight Mile Lake, Interior Alaska, USA).**



### 3.4 Projection for stocks in exchangeable base cations upon permafrost thaw: a new source of nutrients for vegetation

Upon projected permafrost degradation at EML (Garnello et al., 2021), the upper permafrost (0-20 cm below the permafrost table) will undoubtedly thaw and expose soil (total and exchangeable) constituents by 2100. In the upper permafrost layer (0-
20 cm below the permafrost table), we measured stocks in base cations up to 850 g m$^{-3}$ of $Ca_{exch}$, 45 g m$^{-3}$ of $K_{exch}$, 200 g m$^{-3}$ of $Mg_{exch}$ and 150 g m$^{-3}$ of $Na_{exch}$ (Table S5). Upon permafrost thaw, these reservoirs of nutrient base cations initially locked will progressively be released and available for plants and microbial activity.

On average, we observe stocks of exchangeable base cations ($\Sigma\ Ca^{2+}+K^++Mg^{2+}+Na^+$) more than 2-times higher in the
permafrost than in the active layer (~3.7 times for the "organic-thick" permafrost soil profiles; ~2.2 times for the "organic-thin" permafrost soil profiles). The specific stocks of $Ca_{exch}$, $K_{exch}$, $Mg_{exch}$ and $Na_{exch}$ are between 1.5 and 3.5-times higher in permafrost than in the active layer. In particular, the increase in the stock of exchangeable bases is specific to the transition area and is higher between organic and mineral active layers (~2.8-times) than between mineral active layer and permafrost horizons (~1.6-times). The changes in the specific stocks of $Ca_{exch}$, $K_{exch}$, $Mg_{exch}$ and $Na_{exch}$ also differ between the active layer
transition and the permafrost transition (Fig. 9). For instance, stock in $K_{exch}$ decreases from the organic to the mineral horizons of the active layer, but increases within permafrost. In many soil profiles, the increase in $Ca_{exch}$ and $Mg_{exch}$ stocks with depth is more pronounced between the organic and mineral horizons of the active layer than with the permafrost horizons. Overall, these results highlight the increasing reservoir size of available base cations with depth and suggest the release of large amounts of base cations upon permafrost thaw.


The potential release of available nutrient stocks at depth is expected to increase vegetation production and contribute to the ongoing Arctic greening (Keuper et al., 2017; Keyser et al., 2000; Nadelhoffer et al., 1991; Sistla et al., 2013). Moreover, changes in nutrient distribution (higher at depth) may influence the tundra plant species and thereby contribute to the shift in vegetation observed across the Arctic (Chapin et al., 1995; Schuur et al., 2007; van der Kolk et al., 2016). For instance, the
more deeply rooted graminoids access first the newly thawed soil horizons at depth (Hewitt et al., 2019) and thereby benefit first from these newly thawed pools of nutrients. Nevertheless, plant strategies are complex and woody shrubs may also benefit from the deep release of nutrients upon further graminoid nutrient cycling and the nutrient transfer from deep soil horizons to surface litterfall deposition. As changes in Arctic tundra vegetation composition and productivity may generate important feedbacks on climate change (Heijmans et al., 2022), our estimations for stocks in exchangeable cations within permafrost
may be useful for future ecosystem models simulating the evolution of vegetation development and microbial activity upon permafrost thaw (Fisher et al., 2014; Koyama et al., 2014; Sulman et al., 2021; van der Kolk et al., 2016).



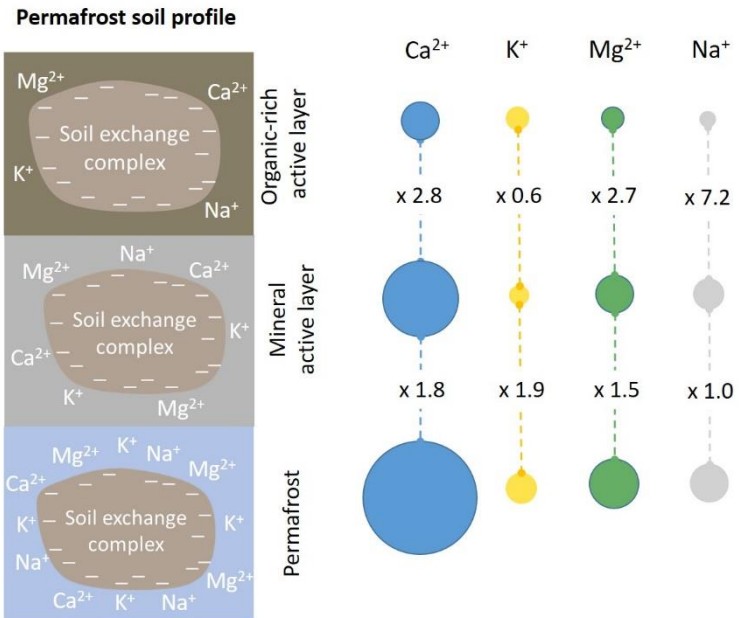

**Figure 9: Schematic representation of the stocks (g m⁻³) in exchangeable base cations (Ca²⁺, K⁺, Mg²⁺, and Na⁺) along a typical permafrost soil profile at the Gradient site (n=85; Eight Mile Lake, Interior Alaska, USA). The average change in stocks with depth is expressed as (i) the ratios between the stock in the organic-rich active layer and the mineral active layer, and (ii) the ratios between the stock in the mineral active layer and the permafrost soil layer. On average, the total stock in exchangeable base cations is about 2-times higher in the permafrost than in the active layer.**

## 4 Conclusion

We characterized the variability with depth of the soil exchange complex properties across a permafrost thaw gradient (constituents, cation exchange capacity, base saturation), and we quantified the stocks in exchangeable cations in the permafrost relative to the seasonally thawed active layer. The main conclusions are:

(i) The base saturation of the soil exchange complex is the highest in permafrost soils (~65%). In the active layer, the base saturation is lower in the organic part (~20%) than in the mineral part (~35%). This reflects the higher exchange acidity in the organic active layers, compared to deeper mineral soil horizons.

(ii) Despite their low OC content and low CEC values, the mineral soil horizons in the active layer and in the permafrost present a higher CEC density (~16 000 cmol$_c$ m⁻³ for mineral active layer; ~12 000 cmol$_c$ m⁻³ for permafrost) than the organic active layer horizons (~9 400 cmol$_c$ m⁻³). This can be explained by the higher bulk density in mineral than in organic soil horizons.

(iii) As a result of the overall increase in base saturation with depth and the overall increase in CEC density with depth, stocks in exchangeable base cations ($Ca^{2+}$, $K^+$, $Mg^{2+}$ and $Na^+$ in $cmol_c$ $m^{-3}$) are more than 2-times higher in the permafrost than in the active layer, with a larger increase in "organic-thick" permafrost soil profiles (~3.7 times) than in "organic-thin" permafrost

soil profiles (~2.2 times).

(iv) The upper permafrost layer (i.e., up to 20 cm below the permafrost table) contains 860 g $m^{-3}$ of $Ca_{exch}$, 45 g $m^{-3}$ of $K_{exch}$, 200 g $m^{-3}$ of $Mg_{exch}$ and 150 g $m^{-3}$ of $Na_{exch}$ as stocks in exchangeable base cations. Given that this upper permafrost is about to thaw in the forthcoming decades, these values are needed to provide constraint on the size of the reservoir in exchangeable nutrients ($Ca^{2+}$, $K^+$, $Mg^{2+}$) for ecosystem models.

This study provides a first order estimate of a reservoir about to thaw and expected to contribute to supply nutrients for vegetation development and microbial activity. The contrast highlighted here in nutrient base cation stocks between the organic and the mineral parts of the active layer and the permafrost about to thaw, is key for changes in tundra vegetation productivity, given that Arctic ecosystems depend on active layer thickness in providing essential nutrients to sustain vegetation growth and development (Iversen et al., 2015; Ping et al., 1998).

## Data availability

All data described in this paper are stored in Dataverse, UCLouvain's Online Repository and accessible through the following DOI: https://doi.org/10.14428/DVN/FQVMEP (Mauclet et al., 2022).

## Authors contribution

EM and SO conceived the project. EM, AM and CH collected the samples in Alaska, USA. EM did the element analysis and

the TOC measurements with the help of AM. MV and EM did the measurements of pH, CEC and exchangeable bases. SO and AM did the mineralogical analyses. EM analyzed the data and calculated to the stocks. EM prepared the manuscript with contributions from all co-authors.

## Competing interests

The authors declare that they have no conflict of interest.

## Acknowledgements

We thank S. Malvaux, M. Gérard, M. Thomas, S. François and the WeThaw team at UCLouvain for valuable scientific exchanges and their help in the field and in the lab. We acknowledge the MOCA analytical platform at UCLouvain for the



analyses: A. Iserentant, C. Givron, L. Monin, and E. Devos. We also thank J. Ledman and the Schuur Lab (Northern Arizona University, Flagstaff) for their scientific support and their help with site selection.

**Funding**

This work was supported by the European Union's Horizon 2020 research and innovation program (grant agreement No. 714617, 2017-2022) and by the Fund for Scientific Research FNRS in Belgium to SO (FC69480).

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
