# Peer review of "Quantifying exchangeable base cations in permafrost: a reserve of nutrients about to thaw"

_Earth System Science Data, 2022_

## Author Comment (AC1)

**Manuscript essd-2022-240: Response to reviewers**

Point-by point response the reviewers' comments with:

**RC : referee comment**

**AC : author comment**

**RC1: 'Comment on essd-2022-240', Matthias Siewert, 21 Sep 2022**

**RC**: The manuscript by Mauclet et al undoubtedly presents an interesting data set on the vertical distribution of soil exchange properties in permafrost soils at a location in Alaska. The authors provide data on CEC, base saturation and other parameters for 7 soil profiles. This data is relevant and useful for the scientific community and provides a valuable reference point to understand the role of permafrost soils in the global carbon cycle. However, the manuscript needs to be thoroughly revised before being published.

AC: We thank the reviewer for the constructive comments which helped improving the manuscript.

**RC**: I believe that the authors should separate between results and discussion. The jump between both and the comparison with the litertaure is not always clear and leading to confusion. In this case, it made it really hard to read the manuscript, filter out key messages and critically evaluate them. The most interesting and relevant discussion point is only in the last paragraph. A couple of things I would be more interested in are: Is there a statistical difference between the organic layer, active layer and upper permafrost (you have this partly) and what is the relevance in an arctic greening context. What nutrients do plants mine? Are there any trends. Further, how could hydrology be relevant for future changes in these systems.

AC: Thanks to all your comments below, we clarified the section "results and discussion". We did statistical tests to compare organic active layer, mineral active layer and permafrost and the test is described at L187-191: "All statistics were performed using R 4.2.1 (RStudio Inc., Boston, Massachusetts, USA; R Core Team, 2018) and plots using the ggplot2 package. The significant differences between organic active layer, mineral active layer and permafrost were evaluated using the non-parametric Kruskal-Wallis test ( $\alpha = 0.05$ ). If significant differences were observed at a p-value <  $\alpha/2$ , a post hoc Dunn's test was carried out.". More specifically, we applied this test throughout the manuscript for Catot, Mgtot, Ktot, Natot, CEC, BS, TRB,  $\frac{Ca_{tot}}{Ca_{exch}}$ ,

 $\frac{Mg_{tot}}{Mg_{exch}}$ ,  $\frac{K_{tot}}{K_{exch}}$ ,  $\frac{Na_{tot}}{Na_{exch}}$ , stock in exchangeable cations and exchangeable cation density. The relevance in an Arctic greening context (with nutrients for plants and hydrology) was addressed in the paragraph L394-412. With these clarifications, we no longer consider that there is a need to separate "results and discussion".

RC: The described correlation (L 206-219) between organic layer and active layer thickness are by no means news. The thermal isolating properties of the OL are well understood. suggest to reduce this result to 1-2 sentences summarizing this and to remove Fig 2 and 3. also This would help streamline the manuscript focus to to on CEC. AC: In order to focus our manuscript on exchangeable base cations, we followed your suggestion and we removed this part and figures associated. We transferred some parts in the section "2.1 Study area and sampling" to explain the choice of the separation of the seven profiles and the link between the active layer depth and the organic-mineral transition at L101-107: "We then divided our seven soil cores according to ALT with the "organic-thin" soil profile (ALT > 60 cm, corresponding to an organic layer thickness OLT < 40 cm) and "organic-thick" soil profile (ALT < 60 cm, corresponding to OLT > 40 cm). This relationship between ALT and OLT can be explained by the low thermal diffusivity and conductivity of the organic matter (Adams, 1973; Farouki, 1981) that insulates the soil (Decharme et al., 2016; Lawrence and Slater, 2008) and thereby influences the permafrost thaw depth."

**RC**: The figures need to be reconsidered. Figure 2 and 3 are irrelevant from a scientific point of view, while figure 7,8 and 9 seem to essentially show the same data in 3 different ways. Also: There are 9 points in Fig 2 but only 7 cores are mentioned in the method section! Are you sure that all results, e.g. L221-225, refer to the mentioned 7 profiles? **AC**: In order to clarify the manuscript, we removed the previous figures 2, 3 that were irrelevant. Moreover, we changed the figure 6 to consider your comments about the stocks and we removed the previous figures 8 and 9.

**RC**: The 7 presented soil profiles are sampled along a gradient that is not explained. There needs to be a better description of the sample location and discussion of the potential soil pedon variability due to pattern ground landforms or environmental gradients. Furthermore, I also would like to see a map of the spatial distribution of the profiles and some sort of indicator reflecting the mentioned gradient, maybe a satellite image. Turbic histic cryosols can show huge variability within a full soil pedon (sensu JL Ping) and the sample location could determine much of the variability of the patterns that you find in your exchange capacity results. This also means a throughout discussion of cryoturbation as a process and explanation for the seen pattern in exchange properties. I am also not sure of the separation between shallow active layer and thick active layer are relevant. It would be much better to express this in terms of soil type or along the mentioned gradient.

AC: We clarified the 7 presented soil profiles sampled throughout the section « 2.1 Study area and sampling". We added a map (Fig. 1) where you can observe the gradient from "MIN" to "EXT" that is explained in the caption: "Figure 1: Study site at Eight Mile Lake, in Central Alaska, USA (inset). The Gradient site is a natural thermokarst gradient originally composed of areas of minimal (Min), moderate (Mod), and extensive (Ext) permafrost thaw. According to the active layer thickness measurement, we separate the seven soil cores into two groups: "organic-thin" soil profile (ALT > 60 cm; in yellow) and "organic-thick" soil profile (ALT < 60 cm; in grey). Source: Esri, HERE, Garmin, OpenStreetMap contributors, and GIS user community.".

In the text, we described better the sample location and the soil profiles separation at L98-107: "A field campaign took place on the natural and monitored gradient of permafrost thaw at EML (Gradient Site; Fig 1; Osterkamp et al., 2009; Vogel et al., 2009) at the late season period between mid-August and early-September in 2019 to sample permafrost soil profiles with contrasted thaw depth from 48 to 88 cm depth. Seven soil cores were collected along the permafrost gradient and their respective active layer thicknesses (ALT) were measured with a metal probe (Table S1). This measurement highlights that ALT no longer follows the gradient of permafrost thaw defined in the literature (Schuur et al., 2021). We then divided our seven soil cores according to ALT with the "organic-thin" soil profile (ALT > 60 cm, corresponding to an organic layer thickness OLT < 40 cm) and "organic-thick" soil profile (ALT < 60 cm, corresponding to OLT > 40 cm). This relationship between ALT and OLT can be explained by the low thermal diffusivity and conductivity of the organic matter (Adams, 1973; Farouki, 1981) that insulates the soil (Decharme et al., 2016; Lawrence and Slater, 2008) and thereby influences the permafrost thaw depth."

There is very limited pedon variability because we studied mineralogy of soil cores in another paper (Mauclet et al., 2023) explained at L107-108: "The mineralogy of the soils from this site is similar and mineral phases observed are plagioclase, K-feldspath, amphibole, quartz, muscovite, vermiculite, kaolinite and illite (Mauclet et al., 2023)."

**RC**: It is unclear if bulk density was measured or only interpolated from previous BD measurements? Please clarify this in L 128ff. If direct BD measurement is still possible on the samples, then this should be the preferred method and added to the manuscript. Then stocks of different parameters were measured, but it is unclear to which depth. Calculation of stocks

should be done to a specific depth for all profiles. I suggest 1m. Otherwise you compare apples with oranges.

AC: Bulk density was estimated from previous measurements, we clarified it in the subtitle L127: "Soil organic carbon content measurement and estimation of soil bulk density". We explain at L135-139: "Given that bulk densities were not measured upon sampling, bulk densities were estimated based on measured SOC content. Because of the strong correlation between the soil organic matter content and the soil bulk density (Askin and Özdemir, 2003; Chaudhari et al., 2013; Périé and Ouimet, 2008), we established a linear regression (Eq. 1; R2 = 0.73) to relate the SOC content (gc kgsoil-1) to soil bulk density (BD, in gsoil cm-3) based on 443 paired measurements on active layer and permafrost samples from a previous study at the EML research site (Interior Alaska, USA; Plaza et al., 2017)."

We calculated the stock in exchangeable base cations (g m-2) up to 1 m depth but we separated between the organic active layer, mineral active layer and permafrost to be able to compare these three different layers. Moreover, to be able compare these three layers independently of their thickness, we calculated the exchangeable base cations density (g m-3). All these changes are well explained in the section "2.3.2 Estimates for CEC density, exchangeable base cations stock and density" at L176-186. We applied these new estimates throughout the manuscript when we talk about density and stock in exchangeable base cations in the sections : "3.2.3 Change in the distribution of exchangeable base cations", "3.3 Influence of vegetation nutrient cycling on the exchangeable base cations density within permafrost soil profiles" and "3.4 Projection exchangeable base cations upon permafrost thaw: a new source of nutrients for vegetation"

**RC**: What was the average weight of the elemental analysis sample? Were loss on ignition measurements (LOI) performed on a larger sample to confirm the representativness of the elemental analysis sample? If not, then you need to discuss this.

**AC**: We specified the methodology of the alkaline fusion at L148-151: "For the fusion, a portion of the ground sample (80 mg) is mixed with lithium metaborate and lithium tetraborate and heated up to 1000°C for 10 minutes. Then, the fusion bead is dissolved in HNO3 2.2 N at 80°C and stirred until complete dissolution. The loss on ignition is assessed at 1000°C, and the total element content is expressed in reference to the dry weight at 105°C."

**RC**: I find the interpretation outlined in L213-219 relating the SOC distribution to permafrost degradation very speculative and not supported. How are these soils more degraded? I assume you don't mean eroded or slope processes, but rather a thermal driven thickening of the active layer. Even differences in the thickness of the active layer may not imply a stronger degradation of the permafrost for these profiles relative to others. Most likely their thermal forcing is rather similar at the scale of the study area. Permafrost soils are highly variable and the seen changes may be observed within a less then a meter for any given soil profile for turbic soils. The difference may also reflect different soil development along the mentioned gradient (that was reduced to two soil types, motivation?). Both would by no means be related to permafrost degradation and even loss of SOC from the system. Most likely they rather reflect inter pedon variability or less SOC accumulation due to environmental gradients. If you have a different opinion on this, then I would expect a detailed chemical or structural analysis of this and a throughout explanation of the mechanisms. Again in L306, I dont think this premise of a significantly reduced organic layer due to thawing is supported by the data shown in the manuscript, the study area description or the cited literature. For instance, Schuur et al 2021 report a loss of 781.6 g C m-2 since the switch to a C source in 1990 and a cumulative projection of 4.18–10.00 kg C m-2 by 2100. Your differences in SOC for individual pedons surpasses potential loss as GHG within a century timescale.

**AC**: As suggested in one of your comment above, we removed this part (previous L213-219) from the manuscript. However, we agree that the term "degradation" is not the best because for us "permafrost degradation" = "permafrost thaw". Thus, we changed along the manuscript the term "permafrost degradation" by "permafrost thaw".

About the gradient and the soil variability, we specified our changes in comments above. We agree about the link between organic layer and permafrost thaw. In our manuscript, we explained that organic layer thickness influences permafrost thaw (L105-107) and we removed L306 that was confusing.

**RC**: Language is overall very good, but would benefit from rephrasing a couple of sentences. e.g. L 25, 31,58-63, 291...

**AC**: We have revised the sentences throughout the manuscript.

**RC**: I hope the authors find these comments helpful and constructive. Iam looking forward to see the results eventually published as I believe that it is a relevant contribution to the field **AC**: We thank the reviewer for the constructive comments.

**RC**: L 14 Which arctic tundra soils are typical?

**AC**: We removed the word "typical" from this sentence.

**RC**: L16 - 'poorly thawed' – never seen this, please use a different expression. How about: permafrost affected soils with a shallow active layer and soils with a thick active layer? AC: As suggested, we simplified the part about OC, so this sentence is not in the abstract anymore. However, we paid attention to remove this expression throughout the manuscript.

**RC**: L 19 - CEC is not defined in the abstract.

**AC**: We added at L12 the definition for CEC and BS: "(cation exchange capacity CEC and base saturation BS)."

**RC**: L24 To what depth did you count the permafrost for the stocks?

**AC**: As mentioned in one comment above, we clarified "stocks" and "density" throughout the manuscript. At this line, we talk about exchangeable base cations density (g m-3), which is here determined by a thickness of 20 cm (20 cm below the permafrost table; L20-22): "More specifically, the exchangeable base cations density in the 20 cm upper part of permafrost about to thaw is ~ 850 g m-3 for Caexch, 45 g m-3 for Kexch, 200 g m-3 for Mgexch and 150 g m-3 for Naexch."

**RC**: L 37 You should mention the word cryoturbation here.

**AC**: We changed the word "cryogenic process" by "cryoturbation". However, we moved this sentence in the section "3.1 Distribution of the organic and mineral constituents from the soil exchange complex within permafrost soil profiles" according to comments of the other reviewer (L199-201): "This supports that the transient layer, that may contain thaw unconformities reflecting previous periods of exceptional permafrost thaw, is sensitive to cryoturbation that can mix soil material by repeated freezing and thawing (Fig. 2; Bockheim and Hinkel, 2007; Ping et al., 2008; Shur et al., 2005)."

**RC**: L 62 Do you mean lateral in the soil as ground water or in streams?

**AC**: We mean the transport from soils to streams, we clarified this sentence at L50-52: "In particular, increased SOM microbial degradation (Hobbie and Chapin, 1998; Nadelhoffer et al., 1992; Schuur et al., 2015; Shaver et al., 2006) or lateral transport of organic soil material from soils to streams (Plaza et al., 2019) generates massive loss of soil organic carbon (SOC)."

**RC**: L 71 I would argue that active layer thickness changes are fairly well understood and quantified.

**AC**: We specify at L61 that the poorly studied part is about nutrient with: "However, these changes in nutrient availability upon permafrost thaw remain poorly quantified..."

**RC**: L79 What do you mean by contrasted? I think you should be more specific here? Do you mean a range of ALT values, or two groups? What are the mean and the SD for these in cm? AC: We understand that this sentence was not clear. We have clarified at L68-72: "Across a range of permafrost soil profiles divided into two groups depending on their active layer thickness (ALT < 60 cm where ALT =  $53 \pm 5$  cm and ALT > 60 cm where ALT =  $76 \pm 12$  cm)...".

**RC**: L87 Which months represent the growing season?

**AC**: We have specified the months at L78-79: "...(for the growing season from May to September; Natali et al., 2012)..."

**RC**: L 88 Your profiles Mod2, Mod3 and EXT3 have less than 35 cm thick organic horizon. The results for SOC also indicate rather different soil types. What soil type are these then? I assume you used the US soil taxonomy system?

AC: We used the WRB classification and specified at L80-81: "Soils at EML are classified as Histic Turbic Cryosols (IUSS Working Group WRB, 2015) and characterized by a thick organic layer up to 55 cm (SOC  $\ge$  20%) in surface."

**RC**: L101 What determined the sample location? How did you chose the gradient? How representative are these locations?

**AC**: As suggested in one of your comment above, we clarified the study site in the section "2.1 Study area and sampling" by explaining the gradient and our choices when sampling.

**RC**: L 102 what are the ranges?

**AC**: We specified the range of the active layer at L100: "...to sample permafrost soil profiles with contrasted thaw depth from 48 to 88 cm depth."

**RC**: L 143 and 146 What do Min1, Mod3, EXT,... stand for?

**AC**: We understand that the name of the soil profiles lacked explanation. As explained above, we added a map (Fig 1.) in the section "2.1 Study area and sampling" where all soil profiles have a name (with explanation in the caption) and an associated group.

**RC**: Table S1, what does n stand for?

**AC**: For each soil profile, we have different samples (n) corresponding to different layers of the soil. We specified this in the caption: "The column "n" corresponds to the number of soil layers sampled for each soil profile."

**RC**: L145 Please motivate the profile selection.

**AC**: We understand that the profile selection throughout the manuscript for each measurement was unclear. We clarified our choices and changed the figures and text accordingly.

For the total concentration in Ca, K, Mg, and Na determined by ICP-OES, we chose two soil profiles that have contrasted permafrost thaw depth, as explained L145-148: "On two selected soil profiles (n = 25) with contrasted permafrost thaw states (one "organic-thick" soil profile Min1 where ALT = 55 cm and one "organic-thin" soil profile Mod3 where ALT = 75 cm) the total concentration in Ca, K, Mg, and Na in bulk soils was determined by inductively coupled plasma optical emission spectroscopy (ICP-OES, iCAP 6500 ThermoFisher Scientific, Waltham, USA) after alkaline fusion (Chao and Sanzolone, 1992)."

In line with the comment from the reviewer, we removed the total concentration in Ca and K determined by pXRF because they do not allow the calculation of the TRB (total reserve in bases) due to the lack of Na and Mg concentrations with this technique.

**RC**: L165 Please motivate the profile selection.

**AC**: The soil mineralogy is available in Mauclet et al 2023, and has been removed from this manuscript. We refer to Mauclet et al 2023 (L108, L220, and L249).

**RC**: L180 Contrasted ALT – The two selected profiles have almost the same ALT 60 and 65 and are the deepest and shallowest in there respective ALT group. Maybe you could give another reason why you selected exactly these two profiles.

AC: In line with the comment from the reviewer to clarify the choice, we decided to focus on the potential CEC and we removed the data of effective CEC.

**RC**: L200 This increase in SOC due to cryoturbation is typical for the bottom of the active layer and top of the permafrost section.

AC: We clarified this by including the paper of Shur et al., 2005 (L197-201): "For some soil profiles, increase in SOC content at the bottom of the active layer (60-80 cm depth; up to 30% OC) may reflect the presence of old organic matter inclusions within the soil mineral phase. This supports that the transient layer, that may contain thaw unconformities reflecting previous periods of exceptional permafrost thaw, is sensitive to cryoturbation that can mix soil material by repeated freezing and thawing (Fig. 2; Bockheim and Hinkel, 2007; Ping et al., 2008; Shur et al., 2005)."

**RC**: Fig 1. It is impossible to distinguish individual profiles. I suggest to differentiate each line by color, or make an average and min/max lines. The lines are also rather thin.

**AC**: We modified this figure (Fig. 2) by doing an average of soil profiles included in the same group. We also added the standard deviation represented by horizontal lines.

**RC**: Fig 4 Any idea why Ca may behave different?

**AC**: We changed the figure 4 (now Fig. 3) to include Min1 ("organic-thick" soil profile) and Mod3 ("organic-thin" soil profile) to show all our data and not only one soil profile. Calcium is the only element that has different concentrations between permafrost and mineral active layer in Min1. However, the trend is not the same in Mod3, so we can't make any conclusion about that.

**RC**: L236 By surface you mean organic? A suggest to statistically compare the organic layer, the mineral layer and the mineral permafrost layer. It would actually be interesting to see if there is a trend across all 7 profiles.

AC: We clarified the layers by talking about organic active layer, mineral organic layer and permafrost in that paragraph (L208-215).

Moreover, we applied a statistical test (as mentioned above) to compare to different layers and to compare here the two soil profiles analyzed (Min1 and Mod3). See changes on Fig. 3 and at L208-215.

**RC**: Fig 5 Again, it is hard to see any trends in this figure.

**AC**: In order to clarify figure 5 (now Fig. 4), as Fig 1, we did an average of soil profiles included in the same group. We also added the standard deviation represented by horizontal lines.

**RC**: Are Fig 7, Fig 8 and Fig 9 essentially showing the same data?

**AC**: We kept the figure 7 (now Fig. 6) but we modified it. We included 4 soil profiles (two "organic-thin" and two "organic-thick" soil profiles) with the exchangeable base cations density (in g m-3) and stock (in g m-2), and the base saturation (in %). Moreover, we merged density and stock in the organic active layer, in the mineral active layer and in the permafrost layer to be able to compare these three different layers.

**RC**: Fig 9 Not sure if the figure is effective at communicating the message. What is the typical profile shallow or thick ALT?

AC: In line with the previous comment, we removed this figure to focus on the now Fig. 6 that is more relevant for the paper.

**RC2: 'Comment on essd-2022-240', Anonymous Referee #2, 14 Dec 2022**

**RC**: The authors quantified the variability with depth of soil exchange complex properties (constituents, cation exchange capacity and base saturation), and the stocks in exchangeable cations in permafrost. The authors found that the CEC density, base saturation, and the average total stock in exchangeable base cations increased with depth. Overall, this dataset is valuable, and the paper is well written. I enjoyed reading this paper.

AC: We thank the reviewer for the constructive comments which helped improving the manuscript.

**RC**: However, my main concern is that although the authors highlighted the importance of the potential reservoir of newly thawed nutrients to promote plant productivity, the permafrost thaw upon warming also increase the potential of the losses of these base cations through leaching. Thus, I think the authors should also highlight this point. Besides, I have some comments though I believe the authors should address in more detail in the results and discussion to make it clear. Please see the following detailed comments.

AC: We understand that leaching can represent a potential loss of base cations. At L312-315, we already highlighted that exchangeable base cations could be leached : "This likely reflects the influence of permafrost thaw on the leaching of exchangeable base cations (Ca2+, K+, Mg2+, and Na+) that are soluble cations. The seasonally thawed mineral active layer is subject to more cation leaching than the upper permafrost (potentially thawed upon exceptional previous warming), and more than the deep permafrost." However, as clarified now at L355-358, we observe that with increasing thaw depth, the density in exchangeable base cations (independent of the thickness) in the mineral active layer is increasing: "This provides newly thawed pool of nutrient base cations that can be sensitive to leaching. There is an increase in exchangeable cations density (independent of the thickness layer) in the mineral active layer at the "organic-thin" (~1100 g m-3) compared to "organic-thick" (~500 g m-3) soil profiles."

**RC**: L21 Please have a brief explanation for the base saturation here.

**AC**: We specify the base saturation at L18-19 with the definition of BS above: "...BS (percentage of CEC occupied by exchangeable base cations Ca2+, K+, Mg2+, and Na+)..."

**RC**: L34–35 This sentence seems redundant in this paragraph. The authors may think about moving it to the place when talking about the transient layer in the section of results and discussion (L327).

AC: We agree that this sentence is redundant in this paragraph. We removed it and added it at L197-201: "For some soil profiles, increase in SOC content at the bottom of the active layer (60-80 cm depth; up to 30% OC) may reflect the presence of old organic matter inclusions within the soil mineral phase. This supports that the transient layer, that may contain thaw unconformities reflecting previous periods of exceptional permafrost thaw, is sensitive to cryoturbation that can mix soil material by repeated freezing and thawing (Fig. 2; Bockheim and Hinkel, 2007; Ping et al., 2008; Shur et al., 2005)."

**RC**: L37–39 I think this sentence can be removed to the section of "Results and discussion" to support the results of increase in SOC content at depth (up to 30% OC; L199–200).

**AC**: We removed this sentence from the introduction and we included it at L200: "...is sensitive to cryoturbation that can mix soil material by repeated freezing and thawing...".

**RC**: L37 Soil acidity sometimes is also one of the mechanisms that leads to high SOM accumulation although low temperature and O2 limitation are the main factors.

AC: We specified that soil acidity also leads to high SOM at L30-33: "Together, cold temperatures, water-saturated conditions, and soil acidity reduce the decomposition rates of

soil organic matter (SOM) mainly originating from dead plant tissues and lead to high SOM accumulation in surface (Schuur et al., 2008; Zimov et al., 2006)."

**RC**: L48–50 This sentence influences the overall flow of this paragraph. Suggest removing it. **AC**: We agree and we removed it.

**RC**: L115 Do you have any data or evidence to show there is no difference in **pH** by using 1:5 proportion vs. 1:15 proportion?

AC: We specified at L123-126 the reason why the comparison between the two methods is still possible because the pH diluted is more acid that the pH 1:5: "Although dilution increases the pH of the soil suspension, regardless of the initial pH value of the soil or the distilled water used in preparation of the suspensions (Peech, 1965), we obtained lower pH in organic soil samples ( $4.0 \pm 0.25$  for pHH2O and  $3.3 \pm 0.29$  for pHKCI) than in mineral soil samples ( $5.0 \pm 0.66$  for pHH2O and  $4.2 \pm 0.66$  for pHKCI)."

**RC**: L148 Could you please provide the details on alkaline fusion method?

**AC**: At L148-151, we added details on alkaline fusion method: "For the fusion, a portion of the ground sample (80 mg) is mixed with lithium metaborate and lithium tetraborate and heated up to 1000°C for 10 minutes. Then, the fusion bead is dissolved in  $HNO_3$  2.2 N at 80°C and stirred until complete dissolution. The loss on ignition is assessed at 1000°C, and the total element content is expressed in reference to the dry weight at 105°C."

**RC**: L155 Please explain the differences in measuring Ca and K in soil by using the nondestructive portable X-ray fluorescence device vs. ICP-OES. Which data did you report in the results?

AC: Following this comment, we removed the total concentration in Ca and K determined by pXRF because total concentration is used to calculate TRB (total reserve in bases, the sum the total concentration in Ca, K, Mg, and Na). The soil profiles added with pXRF don't allow us to determine TRB due to the lack of Na and Mg concentrations.

**RC**: Fig. 2 I was confused about the total number of the points in this figure. Total seven soil cores were sampled. But why were there more than seven points in this figure?

**AC**: As suggested by the other reviewer, we removed this figure. We acknowledge that two other profiles sampled but not analyzed in this study were by mistake on this figure.

**RC**: L214–215 It is unclear here. How can the difference in patterns of SOC distribution between the "organic-thick" and "organic-thin" permafrost soil profiles suggest a potential loss in C with permafrost degradation? Do you mean the organic-thin permafrost soil profiles have a more potential loss in C with permafrost degradation?

AC: We agree with you that this part is unclear and as suggested by the other reviewer, we decided to remove the entire paragraph to stay focus of exchangeable base cations, because the described correlation between organic layer and active layer thickness are by no means news and thermal isolating properties of the organic layer are well understood.

**RC**: L283: Please explain why accumulation of Fe-oxides can increase potential CEC.

AC: We agree that Fe-oxides is not providing negative charges (exchange sites) in the soil conditions (with a soil pH ~4). However, the standard method for the measurement of the potential CEC is with a pH buffered at 7. At pH 7, Fe-oxides can have negative charges and can offer exchange sites for exchangeable cations. We mention at L263-265: "The increase in potential CEC can therefore be explained by a methodologically induced potential CEC associated to negative charges created on Fe-oxides at the pH buffered at 7 in the method (Metson, 1956) but not present in the soil at pH ~4."

**RC**: L315 Please consider change the subtitle as the following paragraphs did not mainly talk about the influence of total reserve in bases on the base saturation.

AC: We agree with you and we changed at L286 by: "Variation of base saturation (BS), pH and total reserve in bases (TRB) along soil profiles"

**RC**: L316 Please explain base saturation here to help remind the readers.

**AC**: At L287-288, we explained BS: "...in the base saturation (BS; percentage of CEC occupied by exchangeable base cations Ca2+, K+, Mg2+, and Na+)..."

**RC**: L320–330 I was confused about the discussion on the overall increase in BS with depth. The authors first referred to another study showing larger concentration of exchangeable K and Ca within permafrost than active layer soils. However, I don't think these results are consistent as the author used the BS in this study to compare with the exchangeable K and Ca in another study. Later, the authors stated that the rare thaw events have likely favored the leaching of the soil base cations of this layer. However, the thaw is less frequent (as mentioned), and then what the mechanism for the leaching was in this transient layers.

AC: We agree with you that we cannot compare BS with concentration in exchangeable cations. This is why we decided to remove this paragraph.

**RC**: L340–341 Based on the above statement, it seems that the data only supported Al3+ and H+ in more acidic soil surface, but not the Ca2+, Mg2+, and K+ in more acidic soil surface unless the authors can show these data.

AC: In order to give a clear message about CEC and to clarify the manuscript, we decided to only focus on potential CEC and not effective CEC. We have removed this paragraph about acid cations.

**RC**: L342–344 Please explain this mechanism. Does the presence of exchangeable acid cations refer to Al3+ as KCI could extract more Al3+ from soil particles than water that caused the lower values of pHKCI than pHH2O?

AC: This concept is well explained in the book from Weil and Brady "Nature and Properties of Soils", 15th edition: the ionic force of KCl is higher than H2O, which leads to the extraction of exchangeable Al3+ that consumes OH- ions and increases H+ concentration. As a result, the solution  $pH_{KCl}$  is lowered.

**RC**: L347 Again, it is worthy to mention here what the total reserve in base to help remind the readers, especially when reading a long paper.

AC: At L305-306, we specified TRB: "... quantified by the total reserve in bases (TRB, sum of total concentration in Ca, K, Mg, and Na; Herbillon, 1986)."

**RC**: L376–378 It seems that the difference in distribution of base cation stocks between the organic-think vs. organic-thin permafrost soils was due to the thickness of the organic layer. Is this right? Or what's the explanation or implication for this difference?

AC: We calculated the stock (g m-2) up to 1 m depth but we separated between the organic active layer, mineral active layer and permafrost to be able to compare these three different layers. Moreover, to be able compare these three layers independently of their thickness, we calculated the exchangeable base cations density (g m-3) that was previously the "stock". All these changes are well explained in the section "2.3.2 Estimates for CEC density, exchangeable base cations stock and density" at L176-186. We applied these new measurements throughout the manuscript when we talk about density and stock in exchangeable base cations : "3.2.3 Change in the distribution of exchangeable base cations density within permafrost soil profiles" and "3.4 Projection exchangeable base cations upon permafrost thaw: a new source of nutrients for vegetation".

At this line, we now talk about the exchangeable base cations density that does not depend on the thickness of the layer and allow us to compare layers.

**RC**: L382–384 Although the permafrost thaw upon warming provides newly thawed pool of nutrient base cations, it also increases their potential of the losses via leaching.

AC: As explained in response to your first comment, we observe that upon thawing, the density in exchangeable base cations (independent of the thickness) in the mineral active layer is increasing (L353-358): "This provides newly thawed pool of nutrient base cations that can be sensitive to leaching. There is an increase in exchangeable cations density (independent of the thickness layer) in the mineral active layer at the "organic-thin" (~1100 g m-3) compared to "organic-thick" (~500 g m-3) soil profiles.". Even if there is leaching, the permafrost thaw provides higher exchangeable base cations.

**RC**: L410–411 I don't think there was enough evidence to support the argument that K is a plant limiting nutrient due to its higher stock in the organic surface horizons than in mineral soil horizons.

AC: At L381, we removed the word "limiting". We did the same mistake at L45 where we also removed the word "limiting".

**RC**: L417 How about the potential of loss for the exchangeable base cations upon permafrost thaw?

AC: As explained in response to your first comment, we observe that upon thawing, the density in exchangeable base cations (independent of the thickness) in the mineral active layer is increasing (L353-358): "This provides newly thawed pool of nutrient base cations that can be sensitive to leaching. There is an increase in exchangeable cations density (independent of the thickness layer) in the mineral active layer at the "organic-thin" (~1100 g m-3) compared to "organic-thick" (~500 g m-3) soil profiles.". Even if there is leaching, the permafrost thaw provides higher exchangeable base cations. Moreover, at L312-315, we already highlighted that exchangeable base cations could be leached : "This likely reflects the influence of permafrost thaw on the leaching of exchangeable base cations (Ca2+, K+, Mg2+, and Na+) that are soluble cations.

---

## Author Response (AR2)

**Manuscript essd-2022-240: Response to reviewers**

Point-by point response the reviewers' comments with:

**RC : referee comment; AR : author response**

**Report #1: 'Comment on essd-2022-240', Anonymous Referee #2, 4 Feb 2023**

**RC:** Thank you for addressing all the comments. The paper is ready to go in my opinion.
**AR:** We thank again the reviewer for his constructive comments.

**Report #2: 'Comment on essd-2022-240', Chaoqun Lu, 6 Jul 2023**

**RC:** Could you please clarify the regression equation used for calculating bulk density? Do you use one single equation (eq. 1) for all three layers, including organic active layers, mineral active layers, and permafrost soil layers? How does this equation perform in explaining the variations of samples in each layer (I suppose $R^2=0.73$ is for 443 samples in total)? If the equation differs in explaining the layer-specific variation, could you discuss its impacts on the stock calculation? In conclusion #2, the higher CEC density in mineral and permafrost layers is attributed to higher bulk density in mineral layer than organic soil layers. Could you provide your opinion about whether and how this one-equation regression (same slope, intercept) affect your conclusion?
**AR:** To answer this question, we have included a new figure (Fig 2) with the regression for the single equation. We have clarified how the equation should be considered: (L139-141) *" This regression reflects the higher bulk density in mineral horizons than in organic horizons (Figure 2), even if the variability in bulk density is higher in soils with low SOC content (mineral horizons)."* Following the suggestion of the reviewer, we have clarified the impact on the stock calculation in the text and in the conclusion #2: (L353-354) *"… considering a higher variability in the estimation for mineral than for organic soils according to the bulk density estimates (Figure 2).";* (L379) *"…even if the variability in bulk density is higher in mineral horizons (Figure 2)";* (L432-433) *"… even if the bulk density estimation is more variable in mineral than in organic horizons."*

**RC:** Please indicate the meaning of error bars in Fig 2, 4, and box plot (middle line, upper, bottom lines and whiskers) in Fig 5.
**AR:** The Fig 2, 4, and 5 from the previous version are now Fig 3, 5 and 6 (in response to the previous comment). We have specified the meaning of the error bars in Fig. 3 and 5: *"The error bars (horizontal lines) represent the standard deviation on the mean values (of 3 to 7 samples)..",* and in Fig. 6: *"In each box plot, the horizontal line represents the median, the end of the box the 25-75% quartiles, and whiskers are 1.5 interquartile ranges from the median. Data points outside of the 1.5 interquartile ranges are represented as dots."*

**Files validated: 'Comment on essd-2022-240', Polina Shvedko, 23 Jan 2023**

**RC:** Your datasets is distributing under CC-BY-NC-ND license. Please note, we accept CC0, CC-BY, ODC-BY, MIT Licence, GNU Generic Public Licence, OGL License. Please consider to change your license. 2. Please add the citation to your DOI https://doi.org/10.14428/DVN/FQVMEP in the "Data availability" of the *.pdf manuscript file. 3. Additionally, please add the full citation for your DOI https://doi.org/10.14428/DVN/FQVMEP into the section "References" of the *.pdf manuscript file.
**AR:** We changed the terms in CC-BY and we added the citation at L451 and L602-604 in the references